# DeepMath - Deep Sequence Models for Premise Selection

**Alexander A. Alemi** *
Google Inc.
alemi@google.com

**François Chollet** *
Google Inc.
fchollet@google.com

**Niklas Een** *
Google Inc.
een@google.com

**Geoffrey Irving** *
Google Inc.
geoffreyi@google.com

**Christian Szegedy** *
Google Inc.
szegedy@google.com

**Josef Urban** *†
Czech Technical University in Prague
josef.urban@gmail.com

## Abstract

We study the effectiveness of neural sequence models for premise selection in automated theorem proving, one of the main bottlenecks in the formalization of mathematics. We propose a two stage approach for this task that yields good results for the premise selection task on the Mizar corpus while avoiding the hand-engineered features of existing state-of-the-art models. To our knowledge, this is the first time deep learning has been applied to theorem proving on a large scale.

## 1 Introduction

Mathematics underpins all scientific disciplines. Machine learning itself rests on measure and probability theory, calculus, linear algebra, functional analysis, and information theory. Complex mathematics underlies computer chips, transit systems, communication systems, and financial infrastructure – thus the correctness of many of these systems can be reduced to mathematical proofs.

Unfortunately, these correctness proofs are often impractical to produce without automation, and present-day computers have only limited ability to assist humans in developing mathematical proofs and formally verifying human proofs. There are two main bottlenecks: (1) lack of automated methods for *semantic* or *formal* parsing of informal mathematical texts (*autoformalization*), and (2) lack of strong automated reasoning methods to fill in the gaps in already formalized human-written proofs.

The two bottlenecks are related. Strong automated reasoning can act as a semantic filter for autoformalization, and successful autoformalization would provide a large corpus of computer-understandable facts, proofs, and theory developments. Such a corpus would serve as both background knowledge to fill in gaps in human-level proofs and as a training set to guide automated reasoning. Such guidance is crucial: exhaustive deductive reasoning tools such as today's resolution/superposition automated theorem provers (ATPs) quickly hit combinatorial explosion, and are unusable when reasoning with a very large number of facts without careful selection [4].

In this work, we focus on the latter bottleneck. We develop deep neural networks that learn from a large repository of manually formalized computer-understandable proofs. We learn the task that is essential for making today's ATPs usable over large formal corpora: the selection of a limited number of most relevant facts for proving a new conjecture. This is known as *premise selection*.

The main contributions of this work are:

- A demonstration for the first time that neural network models are useful for aiding in large scale automated logical reasoning without the need for hand-engineered features.

- The comparison of various network architectures (including convolutional, recurrent and hybrid models) and their effect on premise selection performance.

- A method of semantic-aware "definition"-embeddings for function symbols that improves the generalization of formulas with symbols occurring infrequently. This model outperforms previous approaches.

- Analysis showing that neural network based premise selection methods are complementary to those with hand-engineered features: ensembling with previous results produce superior results.

## 2 Formalization and Theorem Proving

In the last two decades, large corpora of complex mathematical knowledge have been *formalized*: encoded in complete detail so that computers can *fully understand the semantics* of complicated mathematical objects. The process of writing such formal and verifiable theorems, definitions, proofs, and theories is called *Interactive Theorem Proving* (ITP).

The ITP field dates back to 1960s [16] and the Automath system by N.G. de Bruijn [9]. ITP systems include HOL (Light) [15], Isabelle [37], Mizar [13], Coq [7], and ACL2 [23]. The development of ITP has been intertwined with the development of its cousin field of *Automated Theorem Proving* (ATP) [31], where proofs of conjectures are attempted fully automatically. Unlike ATP systems, ITP systems allow human-assisted formalization and proving of theorems that are often beyond the capabilities of the fully automated systems.

Large ITP libraries include the Mizar Mathematical Library (MML) with over 50,000 lemmas, and the core Isabelle, HOL, Coq, and ACL2 libraries with thousands of lemmas. These core libraries are a basis for large projects in formalized mathematics and software and hardware verification. Examples in mathematics include the HOL Light proof of the Kepler conjecture (Flyspeck project) [14], the Coq proofs of the Feit-Thompson theorem [12] and Four Color theorem [11], and the verification of most of the Compendium of Continuous Lattices in Mizar [2]. ITP verification of the seL4 kernel [25] and CompCert compiler [27] show comparable progress in large scale software verification. While these large projects mark a coming of age of formalization, ITP remains labor-intensive. For example, Flyspeck took about 20 person-years, with twice as much for Feit-Thompson. Behind this cost are our two bottlenecks: lack of tools for autoformalization and strong proof automation.

Recently the field of *Automated Reasoning in Large Theories* (ARLT) [35] has developed, including AI/ATP/ITP (AITP) systems called *hammers* that assist ITP formalization [4]. Hammers analyze the full set of theorems and proofs in the ITP libraries, estimate the relevance of each theorem, and apply optimized translations from the ITP logic to simpler ATP formalism. Then they attack new conjectures using the most promising combinations of existing theorems and ATP search strategies. Recent evaluations have proved 40% of all Mizar and Flyspeck theorems fully automatically [20, 21]. However, there is significant room for improvement: with perfect premise selection (a perfect choice of library facts) ATPs can prove at least 56% of Mizar and Flyspeck instead of today's 40% [4]. In the next section we explain the premise selection task and the experimental setting for measuring such improvements.

## 3 Premise Selection, Experimental Setting and Previous Results

Given a formal corpus of facts and proofs expressed in an ATP-compatible format, our task is

**Definition** (Premise selection problem)**.** *Given a large set of premises $\mathcal{P}$, an ATP system $A$ with given resource limits, and a new conjecture $C$, predict those premises from $\mathcal{P}$ that will most likely lead to an automatically constructed proof of $C$ by $A$.*

We use the Mizar Mathematical Library (MML) version 4.181.1147[3] as the formal corpus and E prover [32] version 1.9 as the underlying ATP system. The following list exemplifies a small non-

1147-i386-linux.tar

```
:: t99_jordan: Jordan curve theorem in Mizar
for C being Simple_closed_curve holds C is Jordan;

:: Translation to first order logic
fof(t99_jordan, axiom,  (! [A] :  ( (v1_topreal2(A) & m1_subset_1(A,
k1_zfmisc_1(u1_struct_0(k15_euclid(2)))))  => v1_jordan1(A)) ) ).
```

Figure 1: (top) The final statement of the Mizar formalization of the Jordan curve theorem. (bottom) The translation to first-order logic, using name mangling to ensure uniqueness across the entire corpus.

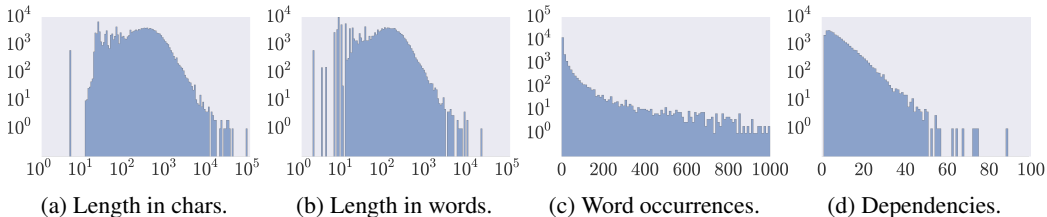

(a) Length in chars.    (b) Length in words.    (c) Word occurrences.    (d) Dependencies.

Figure 2: Histograms of statement lengths, occurrences of each word, and statement dependencies in the Mizar corpus translated to first order logic. The wide length distribution poses difficulties for RNN models and batching, and many rarely occurring words make it important to take definitions of words into account.

representative sample of topics and theorems that are included in the Mizar Mathematical Library: Cauchy-Riemann Differential Equations of Complex Functions, Characterization and Existence of Gröbner Bases, Maximum Network Flow Algorithm by Ford and Fulkerson, Gödel's Completeness Theorem, Brouwer Fixed Point Theorem, Arrow's Impossibility Theorem Borsuk-Ulam Theorem, Dickson's Lemma, Sylow Theorems, Hahn Banach Theorem, The Law of Quadratic Reciprocity, Pepin's Primality Test for Public-Key Cryptography, Ramsey's Theorem.

This version of MML was used for the latest AITP evaluation reported in [21]. There are 57,917 proved Mizar theorems and unnamed top-level lemmas in this MML organized into 1,147 articles. This set is chronologically ordered by the order of articles in MML and by the order of theorems in the articles. Proofs of later theorems can only refer to earlier theorems. This ordering also applies to 88,783 other Mizar formulas (encoding the type system and other automation known to Mizar) used in the problems. The formulas have been translated into first-order logic formulas by the MPTP system [34] (see Figure 1).

Our goal is to automatically prove as many theorems as possible, using at each step all previous theorems and proofs. We can learn from both human proofs and ATP proofs, but previous experiments [26, 20] show that learning only from the ATP proofs is preferable to including human proofs if the set of ATP proofs is sufficiently large. Since for 32,524 (56.2%) of the 57,917 theorems an ATP proof was previously found by a combination of manual and learning-based premise selection [21], we use only these ATP proofs for training.

The 40% success rate from [21] used a portfolio of 14 AITP methods using different learners, ATPs, and numbers of premises. The best single method proved 27.3% of the theorems. Only fast and simple learners such as $k$-nearest-neighbors, naive Bayes, and their ensembles were used, based on hand-crafted features such as the set of (normalized) sub-terms and symbols in each formula.

## 4   Motivation for the use of Deep Learning

Strong premise selection requires models capable of reasoning over mathematical statements, here encoded as variable-length strings of first-order logic. In natural language processing, deep neural networks have proven useful in language modeling [28], text classification [8], sentence pair scoring [3], conversation modeling [36], and question answering [33]. These results have demonstrated the ability of deep networks to extract useful representations from sequential inputs without hand-tuned feature engineering. Neural networks can also mimic some higher-level reasoning on simple algorithmic tasks [38, 18].

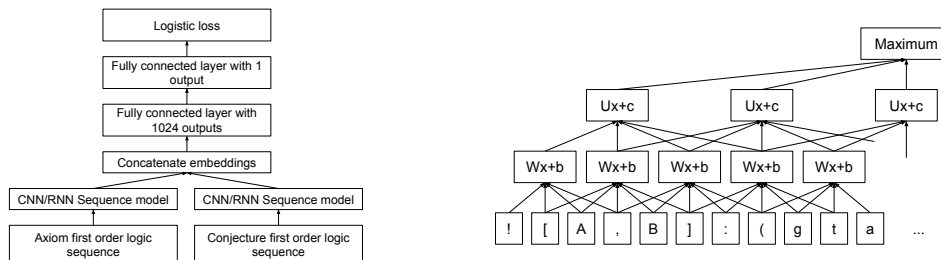

Figure 3: (left) Our network structure. The input sequences are either character-level (section 5.1) or word-level (section 5.2). We use separate models to embed conjecture and axiom, and a logistic layer to predict whether the axiom is useful for proving the conjecture. (right) A convolutional model.

The Mizar data set is also an interesting case study in neural network sequence tasks, as it differs from natural language problems in several ways. It is highly structured with a simple context free grammar – the interesting task occurs only after parsing. The distribution of lengths is wide, ranging from 5 to 84,299 characters with mean 304.5, and from 2 to 21,251 tokens with mean 107.4 (see Figure 2). Fully recurrent models would have to back-propagate through 100s to 1000s of characters or 100s of tokens to embed a whole statement. Finally, there are many rare words – 60.3% of the words occur fewer than 10 times – motivating the definition-aware embeddings in section 5.2.

# 5 Overview of our approach

The full premise selection task takes a conjecture and a set of axioms and chooses a subset of axioms to pass to the ATP. We simplify from subset selection to pairwise relevance by predicting the probability that a given axiom is useful for proving a given conjecture. This approach depends on a relatively sparse dependency graph. Our general architecture is shown in Figure 3(left): the conjecture and axiom sequences are separately embedded into fixed length real vectors, then concatenated and passed to a third network with two fully connected layers and logistic loss. During training time, the two embedding networks and the joined predictor path are trained jointly.

As discussed in section 3, we train our models on premise selection data generated by a combination of various methods, including k-nearest-neighbor search on hand-engineered similarity metrics. We start with a first stage of character-level models, and then build second and later stages of word-level models on top of the results of earlier stages.

## 5.1 Stage 1: Character-level models

We begin by avoiding special purpose engineering by treating formulas on the character-level using an 80 dimensional one-hot encoding of the character sequence. These sequences are passed to a weight shared network for variable length input. For the embedding computation, we have explored the following architectures:

1. Pure recurrent LSTM [17] and GRU [6] networks.
2. A pure multi-layer convolutional network with various numbers of convolutional layers (with strides) followed by a global temporal max-pooling reduction (see Figure 3(right)).
3. A recurrent-convolutional network, that uses convolutional layers to produce a shorter sequence which is processed by a LSTM.

The exact architectures used are specified in the experimental section.

It is computationally prohibitive to compute a large number of (conjecture, axiom) pairs due to the costly embedding phase. Fortunately, our architecture allows caching the embeddings for conjectures and axioms and evaluating the shared portion of the network for a given pair. This makes it practical to consider all pairs during evaluation.

## 5.2 Stage 2: Word-level models

The character-level models are limited to word and structure similarity within the axiom or conjecture being embedded. However, many of the symbols occurring in a formula are defined by formulas

earlier in the corpus, and we can use the axiom-embeddings of those symbols to improve model performance.

Since Mizar is based on first-order set theory, definitions of symbols can be either explicit or implicit. An explicit definition of $x$ sets $x = e$ for some expression $e$, while an implicit definition states a property of the defined object, such as defining a function $f(x)$ by $\forall x. f(f(x)) = g(x)$. To avoid manually encoding the structure of implicit definitions, we embed the entire statement defining a symbol $f$, and then use the stage 1 axiom-embedding corresponding to the whole statement as a word-level embeddings.

Ideally, we would train a single network that embeds statements by recursively expanding and embedding the definitions of the defined symbols. Unfortunately, this recursion would dramatically increase the cost of training since the definition chains can be quite deep. For example, Mizar defines real numbers in terms of non-negative reals, which are defined as Dedekind cuts of non-negative rationals, which are defined as ratios of naturals, etc. As an inexpensive alternative, we reuse the axiom embeddings computed by a previously trained character-level model, mapping each defined symbol to the axiom embedding of its defining statement. Other tokens such as brackets and operators are mapped to fixed pseudo-random vectors of the same dimension.

Since we embed one token at a time ignoring the grammatical structure, our approach does not require a parser: a trivial lexer is implemented in a few lines of Python. With word-level embeddings, we use the same architectures with shorter input sequence to produce axiom and conjecture embeddings for ranking the (conjecture, axiom) pairs. Iterating this approach by using the resulting, stronger axiom embeddings as word embeddings multiple times for additional stages did not yield measurable gains.

## 6 Experiments

### 6.1 Experimental Setup

For training and evaluation we use a subset of 32,524 out of 57,917 theorems that are known to be provable by an ATP given the right set of premises. We split off a random 10% of these (3,124 statements) for testing and validation. Also, we held out 400 statements from the 3,124 for monitoring training progress, as well as for model and checkpoint selection. Final evaluation was done on the remaining 2,724 conjectures. Note that we only held out conjectures, but we trained on all statements as axioms. This is comparable to our k-NN baseline which is also trained on all statements as axioms. The randomized selection of the training and testing sets may also lead to learning from future proofs: a proof $P_j$ of theorem $T_j$ written after theorem $T_i$ may guide the premise selection for $T_i$. However, previous $k$-NN experiments show similar performance between a full 10-fold cross-validation and incremental evaluation as long as chronologically preceding formulas participate in proofs of only later theorems.

### 6.2 Metrics

For each conjecture, our models output a ranking of possible premises. Our primary metric is the number of conjectures proved from the top-$k$ premises, where $k = 16, 32, \ldots, 1024$. This metric can accommodate alternative proofs but is computationally expensive. Therefore we additionally measure the ranking quality using the average maximum relative rank of the testing premise set. Formally, average max relative rank is

$$\text{aMRR} = \operatorname*{mean}_{C} \operatorname*{max}_{P \in \mathcal{P}_{\text{test}}(C)} \frac{\text{rank}(P, \mathcal{P}_{\text{avail}}(C))}{|\mathcal{P}_{\text{avail}}(C)|}$$

where $C$ ranges over conjectures, $\mathcal{P}_{\text{avail}}(C)$ is the set of premises available to prove $C$, $\mathcal{P}_{\text{test}}(C)$ is the set of premises for conjecture $C$ from the test set, and $\text{rank}(P, \mathcal{P}_{\text{avail}}(C))$ is the rank of premise $P$ among the set $\mathcal{P}_{\text{avail}}(C)$ according to the model. The motivation for aMRR is that conjectures are easier to prove if all their dependencies occur early in the ranking.

Since it is too expensive to rank all axioms for a conjecture during continuous evaluation, we approximate our objective. For our holdout set of 400 conjectures, we select all true dependencies $\mathcal{P}_{\text{test}}(C)$ and 128 fixed random false dependencies from $\mathcal{P}_{\text{avail}}(C) - \mathcal{P}_{\text{test}}(C)$ and compute the average max relative rank in this ordering. Note that aMRR is nonzero even if all true dependencies are ordered before false dependencies; the best possible value is 0.051.

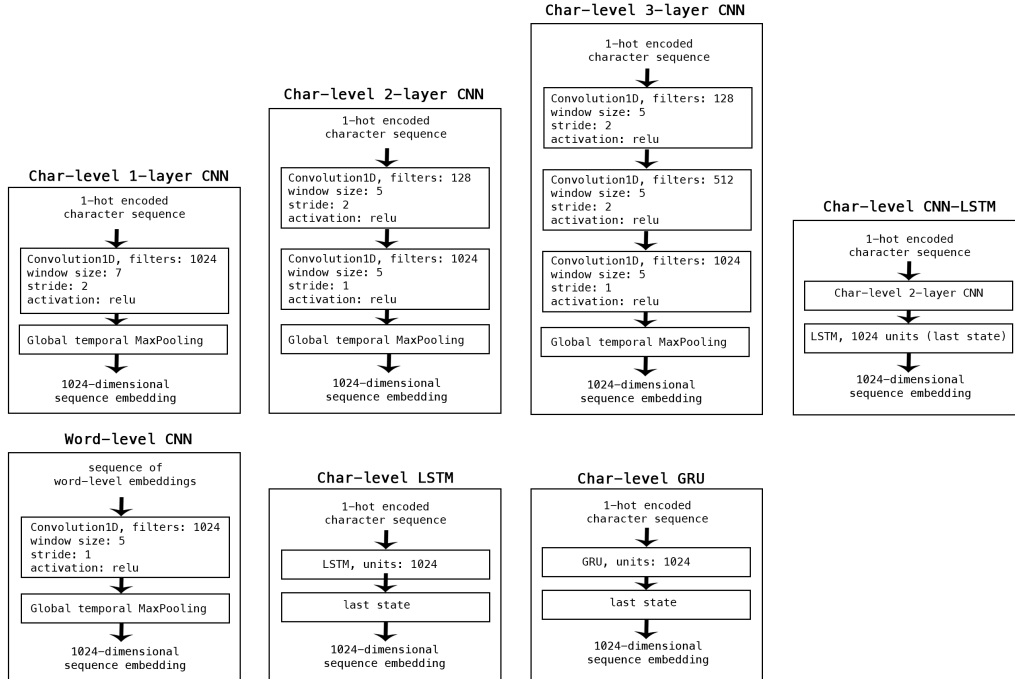

Figure 4: Specification of the different embedder networks.

## 6.3 Network Architectures

All our neural network models use the general architecture from Fig 3: a classifier on top of the concatenated embeddings of an axiom and a conjecture. The same classifier architecture was used for all models: a fully-connected neural network with one hidden layer of size 1024. For each model, the axiom and conjecture embedding networks have the same architecture without sharing weights. The details of the embedding networks are shown in Fig 4.

## 6.4 Network Training

The neural networks were trained using asynchronous distributed stochastic gradient descent using the Adam optimizer [24] with up to 20 parallel NVIDIA K-80 GPU workers per model. We used the TensorFlow framework [1] and the Keras library [5]. The weights were initialized using [10]. Polyak averaging with 0.9999 decay was used for producing the evaluation weights [30]. The character level models were trained with maximum sequence length 2048 characters, where the word-level (and definition embedding) based models had a maximum sequence length of 500 words. For good performance, especially for low cutoff thresholds, it was critical to employ negative mining during training. A side process was continuously evaluating many (conjecture, axiom) pairs. For each conjecture, we pick the lowest scoring statements that have higher score than the lowest scoring true positive. A queue of previously mined negatives is maintained for producing a mixture of examples in which the ratio of mined instances is about 25% and the rest are randomly selected premises. Negative mining was crucial for good quality: at the top-16 cutoff, the number of proved theorems on the test set has doubled. For the union of proof attempts over all cutoff thresholds, the ratio of successful proofs has increased from 61.3% to 66.4% for the best neural model.

## 6.5 Experimental Results

Our best selection pipeline uses a stage-1 character-level convolutional neural network model to produce word-level embeddings for the second stage. The baseline uses distance-weighted $k$-NN [19, 21] with handcrafted semantic features [22]. For all conjectures in our holdout set, we consider all the chronologically preceding statements (lemmas, definitions and axioms) as premise

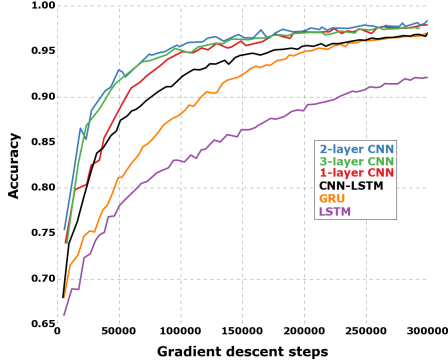

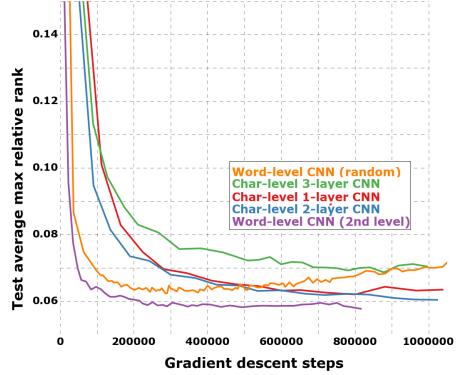

(a) Training accuracy for different character-level models without hard negative mining. Recurrent models seem underperform, while pure convolutional models yield the best results. For each architecture, we trained three models with different random initialization seeds. Only the best runs are shown on this graph; we did not see much variance between runs on the same architecture.

(b) Test average max relative rank for different models without hard negative mining. The best is a word-level CNN using definition embeddings from a character-level 2-layer CNN. An identical word-embedding model with random starting embedding overfits after only 250,000 iterations and underperforms the best character-level model.

candidates. In the DeepMath case, premises were ordered by their logistic scores. E prover was applied to the top-$k$ of the premise-candidates for each of the cutoffs $k \in (16, 32, \ldots, 1024)$ until a proof is found or $k = 1024$ fails. Table 1 reports the number of theorems proved with a cutoff value *at most* the $k$ in the leftmost column. For E prover, we used auto strategy with a soft time limit of 90 seconds, a hard time limit of 120 seconds, a memory limit of 4 GB, and a processed clauses limit of 500,000.

Our most successful models employ simple convolutional networks followed by max pooling (as opposed to recurrent networks like LSTM/GRU), and the two stage definition-based **def-CNN** outperforms the naïve **word-CNN** word embedding significantly. In the latter the word embeddings were learned in a single pass; in the former they are fixed from the stage-1 character-level model. For each architecture (cf. Figure 4) two convolutional layers perform best. Although our models differ significantly from each other, they differ even more from the $k$-NN baseline based on hand-crafted features. The right column of Table 1 shows the result if we average the prediction score of the stage-1 model with that of the definition based stage-2 model. We also experimented with character-based RNN models using shorter sequences: these lagged behind our long-sequence CNN models but performed significantly better than those RNNs trained on longer sequences. This suggest that RNNs could be improved by more sophisticated optimization techniques such as curriculum learning.

| Cutoff | $k$-NN Baseline (%) | char-CNN (%) | word-CNN (%) | def-CNN-LSTM (%) | def-CNN (%) | def+char-CNN (%) |
|---|---|---|---|---|---|---|
| 16 | 674 (24.6) | 687 (25.1) | 709 (25.9) | 644 (23.5) | 734 (26.8) | **835 (30.5)** |
| 32 | 1081 (39.4) | 1028 (37.5) | 1063 (38.8) | 924 (33.7) | 1093 (39.9) | **1218 (44.4)** |
| 64 | 1399 (51) | 1295 (47.2) | 1355 (49.4) | 1196 (43.6) | 1381 (50.4) | **1470 (53.6)** |
| 128 | 1612 (58.8) | 1534 (55.9) | 1552 (56.6) | 1401 (51.1) | 1617 (59) | **1695 (61.8)** |
| 256 | 1709 (62.3) | 1656 (60.4) | 1635 (59.6) | 1519 (55.4) | 1708 (62.3) | **1780 (64.9)** |
| 512 | 1762 (64.3) | 1711 (62.4) | 1712 (62.4) | 1593 (58.1) | 1780 (64.9) | **1830 (66.7)** |
| 1024 | 1786 (65.1) | 1762 (64.3) | 1755 (64) | 1647 (60.1) | 1822 (66.4) | **1862 (67.9)** |

Table 1: Results of ATP premise selection experiments with hard negative mining on a test set of 2,742 theorems. Each entry is the number (%) of theorems proved by E prover using that particular model to rank the premises. The union of def-CNN and char-CNN proves 69.8% of the test set, while the union of the def-CNN and k-NN proves 74.25%. This means that the neural network predictions are more complementary to the k-NN predictions than to other neural models. The union of all methods proves 2218 theorems (80.9%) and just the neural models prove 2151 (78.4%).

# 7 Conclusions

In this work we provide evidence that even simple neural models can compete with hand-engineered features for premise selection, helping to find many new proofs. This translates to real gains in

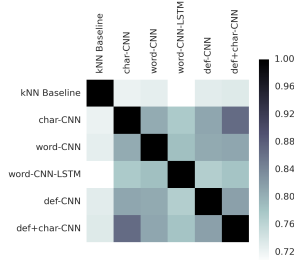

(a) Jaccard similarities between proved sets of conjectures across models. Each of the neural network model prediction are more like each other than those of the $k$-NN baseline.

| Model | Test min average relative rank |
|---|---|
| char-CNN | 0.0585 |
| word-CNN | 0.06 |
| def-CNN-LSTM | 0.0605 |
| def-CNN | **0.0575** |

(b) Best sustained test results obtained by the above models. Lower values are better. This was monitored continuously during training on a holdout set with 400 theorems, using all true positive premises and 128 randomly selected negatives. In this setup, the lowest attainable max average relative rank with perfect predictions is $0.051$.

automatic theorem proving. Despite these encouraging results, our models are relatively shallow networks with inherent limitations to representational power and are incapable of capturing high level properties of mathematical statements. We believe theorem proving is a challenging and important domain for deep learning methods, and that more sophisticated optimization techniques and training methodologies will prove more useful than in less structured domains.

# 8 Acknowledgments

We would like to thank Cezary Kaliszyk for providing us with an improved baseline model. Also many thanks go to the Google Brain team for their generous help with the training infrastructure. We would like to thank Quoc Le for useful discussions on the topic and to Sergio Guadarrama for his help with TensorFlow-slim.

## Footnotes

*Authors listed alphabetically. All contributions are considered equal.

†Supported by ERC Consolidator grant nr. 649043 *AI4REASON*.

[3] ftp://mizar.uwb.edu.pl/pub/system/i386-linux/mizar-7.13.01_4.181.

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
