[Reviews · NeurIPS 2016]

Reviewer 1

Summary

This paper considers the problem of premise selection for automated theorem proving: given a set of axioms (premises) and a conjecture, select a subset of axioms to provide the theorem prover that will allow proving the conjecture. The approach is to embed axioms and conjectures into real-valued vectors, and then map the concatenation of embeddings to a classification of whether the axiom is relevant for proving the conjecture. A variety of neural network architectures are explored for representing the axioms and conjectures, including recurrent and convolutional neural network approaches. In addition, a two-stage approach is developed, where embeddings for identifiers are created by embedding the identifier’s definition. Results show that when a small number of premises are returned (up to 64), a k-NN baseline performs best. When up to 1024 premises are returned, one of proposed models proves 62.95% of conjectures versus the k-NN baseline’s 59.34%.

Qualitative Assessment

Pros: * The problem is interesting and does not appear to have previously received attention from the NIPS community * The paper is clearly written Cons: * The approach is not particularly novel; it is mostly a case of applying a range of existing neural network components to the premise selection problem. * The improvements achieved over a k-NN baseline are small, and I have some reservations about how the experimental results are reported (see below) Experiments * Numbers are presented for the performance of previous work in Section 3, but they don’t appear to be comparable to the results reported in the experiments (Section 6). Is there a way to meaningfully compare these numbers? If not, it's confusing to include them. * Table 1 is not a fair way of reporting results. This presentation is asking us to take a maximum of test performance over all the different neural network variants, and to compare that to the performance of a single baseline method. However, there is quite a lot of variance in the neural network performance relative to the difference between neural network and baseline performance. Thus, there is a bias towards selecting one of the neural networks as the best method, even though it could just be due to the fact that we’re taking a max over many noisy performances. In section 3 it is mentioned that a portfolio of simple learners significantly outperform any single learner. If such an extensive search is being done over tweaks to the proposed method, then an equally extensive search should be done over baseline methods. Presentation * Please number Figures 5 and 6 * Figure 5 isn’t readable in black and white * I find the title off-putting. “DeepMath” is a very general term relative to what the paper actually does (premise selection for automated theorem proving). It sounds like a title meant to grab headlines as opposed to being descriptive of the scientific contributions.

Confidence in this Review

2-Confident (read it all; understood it all reasonably well)


Reviewer 2

Summary

This paper provides a novel application of deep learning to automated theorem provers by training convolutional neural networks to perform premise selection.

Qualitative Assessment

The paper is well written and provides suitable background material to motivate the problem for ML practitioners who are not intimately familiar with ITP/ATP. The formulation of premise selection as a machine learning problem is clear and concise. The fact that the techniques proposed are validated against a popular, public data set (Mizar) is a strength. Sections 5.1 and 5.2 would benefit from more explicit examples of how the character and word models are produced from the input data. An additional figure would likely be sufficient.

Confidence in this Review

2-Confident (read it all; understood it all reasonably well)


Reviewer 3

Summary

The authors experimented with various deep learning architectures to select premises for an automated theorem prover (E prover) to prove theorems. Their two stage method consists of (1) character-level models, and (2) word-level models where (1) avoids any hand engineered character level features and (2) helps with semantic understanding of higher level mathematical definitions. Their experiments compared (a) k-NN lookup baseline with (b) LSTM/GRU sequence networks and (c) various combinations of char-word level CNN architectures. Their results provided evidence that their CNN networks are competitive with the hand-engineered featured bootstrapped k-NN baseline method and was able to find new proofs not previously available. The architectures presented are simple and nascent providing pointers for other in further developing in neural network based Premise Selection in the ATP area.

Qualitative Assessment

There seems to be a disconnect between motivating the work by interesting challenges posed to RNNs (distribution lengths) and the lack of analyses about why the simple CNNs performed way better than the RNN models. The application is interesting, the network architectures are simple/clear, and the main contribution is the ability for the CNNs to capture hand engineered feature in previous baseline. This looks like a good poster for NIPS but more depth on RNN/CNN comparison needed for a more impressive entry.

Confidence in this Review

2-Confident (read it all; understood it all reasonably well)


Reviewer 4

Summary

In this paper, several deep sequence model with different network architectures are proposed for premise selection (an essential procedure in automatic theorem proving). The proposed definition-based model which relies on the low level character embeddings significantly improves the performance. This work is interesting and innovative since this is the first one to incorporate the deep learning model into automatic theorem proving and the results also outperforms the conventional approach. But some details seem not well explained.

Qualitative Assessment

This work is supposed to be the first trier of incorporating the deep learning model into automatic theorem proving. The best of the 5 neural network model, def+char-CNN outperforms the k-NN baseline when cutoff is more than 128, but the 5 proposed networks all fall behind the baseline when cutoff is lower than 128. The authors may provide an explanation. Technical comments: 1. According to figure 2, the length of statements has a wide range with at most ten thousands of characters. But in Line 203-205, it says the length of input sequence for the character level model is at most 2048, and the length for word level model is at most 500. So what if the length of a statement (axiom or conjecture) has more than 2048 characters or more than 500 words? And how do the different embedder networks handle the variation of length of the input sequences? This is not clearly explained. 2. The one-hot encoding is used for representing the 80 characters, i.e. for each character there will be 79 zeros and only 1 one. Then most of the input of the character level network will be zero. And the window size of convolution filter is no more than 5. Intuitively, this will not work for a convolutional neural networks. The authors should provide more info.

Confidence in this Review

2-Confident (read it all; understood it all reasonably well)


Reviewer 5

Summary

These are two of the key problems in automated theorem proving: 1. Parsing the theorem -- e.g. (ASCII text or a syntax tree) --> (appropriate internal representation/featurization) 2. Premise selection -- "Filling in the gaps" of what isn't explicitly mentioned by the humans who formulated the problem. This paper tackles problem (1) by architecting a neural network that takes plain-text input (an Axiom), and feeds it to hidden layers of which at least one is convolutional and can produce a fixed-size output. Addressing problem (2), the neural net outputs a Conjecture, which if I understand correctly is correct information that wasn't explicitly provided in the original problem. (I think this is right. Maybe the conjecture is the proof itself? But, I guess the goal is to solve problem (2) above.)

Qualitative Assessment

The work itself is quite interesting. I would like to see the authors clarify the following. 1. *More clear motivation.* I voted "sub-standard" for potential impact and usefulness. The authors are welcome to "prove" me wrong! But, as it stands, the paper doesn't have much of a motivation section. What are the "great unknown" proofs that we hope that machine learning can solve? 2. *Output form.* What does the output of the neural network look like? Is there truly only one correct answer (which is presumably in the test set)? Is it outputting a snippet of text? A binary "yes or no?" 3. *Appropriate baselines.* In Section 2, the authors mention several previous approaches to automated theorem proving (e.g. HOL, Coq, etc). The authors are training and testing on a widely-available dataset. However, if I understand correctly, all entries in the results table are generated by the authors. I'd like to see the following: - A baseline from the related work. (I'm somewhat flexible on how this is organized. It could be "code or algorithm from the Theorem Proving related work, run by you on the appropriate dataset." Or, it could be "number from related work's results table, placed into the context in this paper.") - A "random" baseline. Is this binary classification where "random" = 50% accuracy? 4. *Qualitative Results.* The authors provide quantitive results in the form of "percent of proofs proven correctly." How about some example proofs that the algorithms got correct/incorrect? 5. *Clarity on where the metric came from.* The authors provide a metric in Section 6.2 for evaluating the results. Is this a standard metric? If so, cite the source (and point to other authors' results on this dataset scored using this metric). If not, claim your new metric and explain why this metric is important!

Confidence in this Review

2-Confident (read it all; understood it all reasonably well)